# Prognostic and Therapeutic Implications of Immune Classification by CD8^+^ Tumor-Infiltrating Lymphocytes and PD-L1 Expression in Sinonasal Squamous Cell Carcinoma

**DOI:** 10.3390/ijms22136926

**Published:** 2021-06-28

**Authors:** Rocío García-Marín, Sara Reda, Cristina Riobello, Virginia N. Cabal, Laura Suárez-Fernández, Blanca Vivanco, César Álvarez-Marcos, Fernando López, José L. Llorente, Mario A. Hermsen

**Affiliations:** 1Department Head and Neck Oncology, Instituto de Investigación Sanitaria del Principado de Asturias, 33011 Oviedo, Spain; rociogm220879@hotmail.com (R.G.-M.); cristinarisu15@gmail.com (C.R.); vircabal@hotmail.com (V.N.C.); laura_quillo@hotmail.com (L.S.-F.); 2Department Otolaryngology, Hospital Universitario Central de Asturias, 33011 Oviedo, Spain; sara_reda_91@hotmail.com (S.R.); caalvarez@uniovi.es (C.Á.-M.); flopez_1981@yahoo.es (F.L.); llorentejlx@gmail.com (J.L.L.); 3Department Pathology, Hospital Universitario Central de Asturias, 33011 Oviedo, Spain; vivancoblanca@gmail.com

**Keywords:** sinonasal cancer, squamous cell carcinoma, CD8^+^ TILs, PD-L1, immunotherapy

## Abstract

Sinonasal squamous cell carcinoma (SNSCC) is an aggressive tumor predominantly arising in the maxillary sinus and nasal cavities. Advances in imaging, surgical and radiotherapeutic techniques have reduced complications and morbidity; however, the prognosis generally remains poor, with an overall 5-year survival rate of 30–50%. As immunotherapy may be a new therapeutic option, we analyzed CD8^+^ tumor-infiltrating lymphocytes (TILs) and the tumor microenvironment immune type (TMIT, combining CD8^+^ TILs and PD-L1) in a series of 57 SNSCCs. Using immunohistochemistry, tissue samples of 57 SNSCCs were analyzed for expression of CD8 on TILs and of PD-L1 on tumor cells. The results were correlated to the clinical and survival data. In total, 88% (50/57) of the tumors had intratumoral CD8^+^ TILs; 19% (11/57)—CD8^high^ (>10%); and 39/57 (68%)—CD8^low^ (1–10%). PD-L1 positivity (>5%) was observed in 46% (26/57) of the SNSCCs and significantly co-occurred with CD8^+^ TILs (*p* = 0.000). Using univariate analysis, high intratumoral CD8^+^ TILs and TMIT I (CD8^high^/PD-L1^pos^) correlated with a worse survival rate. These results indicate that SNSCCs are immunogenic tumors, similar to head and neck squamous cell carcinomas. Nineteen percent of the cases were both CD8^high^ and PD-L1^pos^ and this subgroup may benefit from therapy with immune checkpoint inhibitors.

## 1. Introduction

Malignant tumors of the sinonasal cavity are rare, accounting for less than 5% of all head and neck cancers [1]. Sinonasal squamous cell carcinoma (SNSCC) is the most common subtype of epithelial tumors of the sinonasal cavity which predominantly occurs in the maxillary sinus and the nasal cavity and accounts for 50–80% of all sinonasal malignances, with the highest incidences reported in European countries [2].

Partly due to the nonspecific early clinical symptoms prognosis, patients are often diagnosed late and with advanced-stage tumors. Treatment includes surgery with adjuvant radiotherapy or a combination of radiotherapy and chemotherapy [3,4]. Advances in imaging techniques, endoscopic surgical approaches and precision radiotherapy have reduced complications and morbidity and improved patient recovery. However, the prognosis generally remains poor, with an overall 5-year survival rate of 30–50% with frequent local recurrence as the main contributor to mortality [2,5]. It is clear that new treatment options are needed for SNSCCs, and targeted immunotherapy may be such an option.

Treatment with immune checkpoint inhibitors improves outcomes for several different malignancies, including advanced metastatic cancers that previously had minimal hope for long-term survival [6]. Tumor cells are able to evade the immune system by modulating immune checkpoint pathways, as well as by increasing programmed death-ligand 1 (PD-L1), making tumor-infiltrated lymphocytes (TILs) that express the programmed cell death protein 1 (PD-1) receptor ineffective despite being present in the tumor microenvironment [7,8]. Clinical trials using immunotherapy have led to the approval of anti-PD-1 agents such as pembrolizumab and nivolumab for a number of tumor types including head and neck squamous cell carcinomas (HNSCC) [9,10,11].

Proposed predictive biomarkers for the efficacy of immunotherapy include the presence of TILs, PD-L1 expression and tumor mutational burden [12,13]. Previous studies reported 30–34% PD-L1 expression in SNSCCs, indicating a subgroup of patients that may benefit from immunotherapy [14,15]. The presence of CD8^+^ TILs has been associated with favorable clinical outcomes in patients with HNSCCs [15,16,17,18,19,20,21] and also in sinonasal tumors including SNSCCs, intestinal type adenocarcinomas (ITAC) and olfactory neuroblastomas (ONB) [15,22,23]. Tumor microenvironment immune type (TMIT) is a classification recognizing four immunotypes based on the presence of intratumoral CD8^+^ TILs and tumoral PD-L1 expression [24,25]. The clinical relevance of this classification is supported by its correlation with high mutational burden and oncogenic viral infection [25] and may predict the response to immunotherapy on an individual basis [26,27,28]. We studied both CD8^+^ TILs and TMIT in a series of 57 SNSCCs as possible predictive markers for immunotherapy. In addition, the results were correlated with the clinical and survival data.

## 2. Results

### 2.1. Clinical Features and Follow-Up

Forty-one of the 57 (72%) SNSCC patients received radiotherapy after radical surgery. The median follow-up was 18 months (range, 1–312). Twenty (35%) patients were female and 37 (65%) were male; 86% of the tumors originated in the maxillary sinus. The predominant disease stage was IVa (44%), followed by stages III (30%) and IVb (19%). The differentiation grade was good in 39% of the patients; moderate—in 23%; poor—in 39%. The mean follow-up was 39 months (range, 1–312). During this period, 44/57 (77%) patients developed local recurrence, 7/57 (12%)—metastasis. At the time of writing, eight (14%) patients remained alive, 41 (72%) died of the disease and eight died of other causes. The 5-year disease-specific survival was 25%. Details of all the clinical characteristics are presented in Table 1.

### 2.2. CD8^+^ TILs and Correlation with the Clinical and Survival Data

Stromal CD8^+^ TILs were detected in almost all the patients (98.2%, 56/57), being low in 56.1% (32/57) of the patients; high—in 42.1% (24/57); absent—in 1.8% (1/57). Intratumoral CD8^+^ TILs were present in 87.7% (50/57) of the cases; evaluated as absent in 12.3% (7/57) of the cases; low—in 68.4% (39/57); and high—in 19.3% (11/57) (Table 1, Figure 1). Neither intratumoral nor stromal presence of CD8^+^ TILs correlated with clinical parameters, including grade of differentiation, disease stage and the development of recurrences or metastases. Nevertheless, we observed that 100% of the patients with a high intratumoral CD8^+^ expression level developed recurrence and 91% died of the disease (Table 1). Univariate Kaplan–Meier survival analysis demonstrated a significantly worse overall and disease-specific survival (logrank *p* = 0.023 and 0.006, respectively) in the cases where intratumoral CD8^+^ TILs were detected (Figure 2). Additional univariate analysis of only the 41 patients who had received radiotherapy yielded an even stronger correlation between intratumoral CD8^+^ TILs and the worse overall and disease-specific survival (logrank *p* = 0.002 and 0.003, respectively, Figure 3). The presence of CD8^+^ TILs in the stromal compartment did not correlate either with clinical features or overall or disease-specific survival (logrank *p* = 0.418 and 0.106, respectively).

### 2.3. TMIT and Correlation with the Clinical and Survival Data

PD-L1 expression in >5% of the tumor cells was observed in 26/57 (46%) SNSCCs. Combining the presence of intratumoral CD8^+^ TILs and of PD-L1 on tumor cells, we distinguished between the following TMITs: type I—11/57 (19%); type II—31/57 (54%); and type III—15/57 (26%) (Table 1, Figure 1). TMIT IV was not observed as all the CD8^high^ cases also expressed PD-L1. The TMIT was not significantly related to clinicopathological parameters such as disease stage, degree of differentiation or the development of recurrence or metastasis (Table 1). TMIT II and, to a lesser extent, type III demonstrated a significantly better overall (logrank *p* = 0.036) and disease-specific survival (logrank *p* = 0.025) in the univariate survival analysis. Notably, all the TMIT I patients died of the disease within 5 years, whereas the type II and III patients featured a 5-year disease-specific survival of 24% and 48%, respectively (Figure 2). Additional univariate analysis of only the 41 patients that had received radiotherapy showed an even stronger correlation between the TMIT and the overall and disease-specific survival (logrank *p* = 0.003 and 0.012, respectively, Figure 3).

### 2.4. CD8^+^ TILs Co-Occurs with Tumoral PD-L1-Positive Expression

The presence of intratumoral CD8^+^ TILs correlated significantly with PD-L1 expression by tumor cells: PD-L1 positivity was seen in 100% of the CD8^high^ tumors, in 36% of the CD8^low^ tumors and in 14% of the CD8^−^ tumors (Table 2, Pearson’s chi-squared test, *p* = 0.000).

## 3. Discussion

Currently, treatment of SNSCCs consists of open or endoscopic surgery combined with radiotherapy, sometimes with complementary chemotherapy [5]. Tumors with a generally aggressive clinical behavior being rare, there is an unmet clinical need for alternative treatment options. Here, we presented data on CD8^+^ TILs and PD-L1 positivity as indicators for immunotherapy in these tumors.

Approximately 88% of the cases displayed presence of CD8^+^ TILs (19%—high; 69%—low) in the intratumoral compartment, while all the tumors but one also featured stromal presence. This result suggests that SNSCCs are immunogenic tumors and is in agreement with the previously published studies on SNSCCs and HNSCCs reporting CD8^+^ TILs in 35–50% of cases [15,16,17,18,19,20,21]. In contrast, in ITACs and low-grade ONBs, two other tumor types of the sinonasal region, CD8^+^ TILs were much less frequent and were predominantly found in the stromal compartment [22,23,29].

Using univariate analysis, we further observed that the presence of CD8^+^ TILs significantly correlated with worse survival, with the CD8^high^ cases even more so than the CD8^low^ ones (Figure 2). This correlation was only significant with the intratumoral presence and not with the stromal presence, suggesting that the actual contact between tumor cells and CD8^+^ TILs is important for the prognostic value. Classifying our series of SNSCCs according to the TMIT did not yield additional prognostic information; TMIT I (CD8^high^) presented worse survival compared to TMITs II and III (both CD8^low^). This result contrasts with reports on other sinonasal tumors, including SNSCCs, ITACs, ONBs [15,21,22,23,29] and histologically similar HNSCCs [17,18,19,25]. In fact, a positive relation between high CD8^+^ TIL scores and favorable prognosis is found in the majority of cancer types such as ovarian [30], renal [31], lung [32] and pancreatic cancers [33], although a number of studies have found high CD8^+^ TIL scores related to worse survival [34,35,36]. As our study is retrospective with a low number of heterogeneous tumors precluding multivariate survival analysis, our data must be considered with caution. Discrepancies may be related to differences in patient cohorts and treatment schemes. In contrast to other reports, our series consisted of a relatively high proportion (63%) of cases with disease stage IV while radiotherapy was administered in only 72% of the patients and none received chemotherapy. To take into consideration the influence of radiotherapy on survival, we carried out a separate univariate survival analysis of only the 41 patients that had received radiotherapy after radical surgery. This produced an even stronger correlation of CD8^+^ TILs and TMIT I with worse overall and disease-specific survival (Figure 3) and therefore does not offer an explanation for the discrepancy with the main body of literature. Tumor cells are thought to avoid cytolysis by activated T cells by means of expressing PD-L1. Thus, the combination of tumoral PD-L1 expression and low CD8^+^ TIL counts (TMIT III) should be associated with evasion of the immune system and a worse clinical course. Nevertheless, we found exactly the opposite. Observing that all the tumors with high CD8^+^ TIL scores in our series were also PD-L1^pos^ (Table 2), we speculate that CD8^+^ TILs are dysfunctional exhausted cells. Another possible explanation may be sought in the tumoral HLA class I expression levels. Decreased or zero expression of the HLA class I antigens caused by viral infection or (epi)genetic regulation is frequently observed in solid tumors and can inhibit CD8^+^ TILs to recognize tumor cells [37,38]. Human papillomavirus (HPV), indeed, plays a role in SNSCCs. Analyzed as part of a previously published study by our group, 10% (5/50) of our cases were found positive [39], similar to other SNSCC studies [40,41,42]. These HPV-positive cases may be thought to escape immune surveillance.

Aside from the prognostic value, CD8^+^ TILs and TMIT classification are considered predictors for the efficacy of PD-L1/PD-1 immune checkpoint inhibitors. Even if CD8^+^ TILs and PD-L1 expression are associated with worse survival, as our data indicate, SNSCCs appear to be immunogenic tumors where immunotherapy may present a new possibility for the treatment of this aggressive type of cancer. In locally advanced HNSCCs, clinical trials have shown promising results with PD-1 inhibitors. However, there is a need for predictive biomarkers as clinical responses have been observed in only 13–22% of HNSCC patients [9,10,43]. For immunotherapy to be effective, tumor-activated CD8^+^ TILs need to be present in PD-L1-expressing tumors [28]. These two factors are reflected in TMIT I (CD8^high^/PD-L1^pos^), which we observed in 19% of the SNSCCs, very similar to the 23% found by Quan et al. [20]. This proportion is lower than in highly immunogenic tumors such as melanomas, renal cell and bladder cancers, HNSCCs or lung squamous cell carcinomas (40–50% of the cases belong to TMIT I) [25], but still higher than in other sinonasal cancers such as ITACs and ONBs [22,23,29].

In conclusion, immune checkpoint inhibitors are emerging as new options for treatment of many tumor types. Pembrolizumab and nivolumab have already been approved by the FDA and the EMA for recurrent and metastatic HNSCCs and may also be considered for recurrent SNSCCs. Indeed, SNSCCs are aggressive tumors with frequent recurrence for which very few therapeutic alternatives are available. Our data revealed 19% of SNSCCs with high presence of CD8^+^ TILs and PD-L1 positivity (TMIT I), and this subgroup might benefit from therapy with immune checkpoint inhibitors.

## 4. Materials and Methods

### 4.1. Patients and Specimens

The primary tumor samples were obtained from 57 previously untreated SNSCC patients at the Department of Otolaryngology at Hospital Universitario Central de Asturias (Oviedo, Spain). Written informed consent for the collection, storage and analysis of specimens was obtained from all the patients. The study had received prior approval from our institutional ethical committee (approval number 07/16 for project CICPF16008HERM, 11 November 2017).

### 4.2. Immunohistochemistry

Tissue microarray (TMA) blocks were prepared from formalin-fixed paraffin-embedded tumor tissues using a Beecher Tissue Microarrayer (Beecher Instruments, Silver Spring, MD, USA). In total, four TMA blocks were constructed, containing three 1-mm cores from different areas of 57 sinonasal tumors. Each block included normal sinonasal mucosa adjacent to the tumor tissue samples as the internal control. Three-micrometer sections were stained with hematoxylin and eosin and reviewed by one pathologist to determine whether the samples featured a good representation of the original tumor blocks. Immunohistochemistry (IHQ) was performed on an automatic staining workstation (Dako Autostainer Plus; DakoCytomation, Glostrup, Denmark). Antibodies anti-PD-L1 (clone E1L3N) (1/100 monoclonal rabbit antibodies, Cell Signaling Technology, Cambridge, UK), anti-CD8 (clone C8/144B), IR623 (prediluted monoclonal mouse antibodies, DAKO, Glostrup, Denmark) were applied using high-pH antigen retrieval for 20 min. The slides were evaluated by three observers (C.R., S.R. and B.V.); in those samples where there was discrepancy, ot was solved afterwards by looking together using a multi-head microscope. The staining was visualized by light microscopy.

The density of CD8^+^ TILs was scored in a semi-quantitative manner as 0% (negative), 1–10% (low) and >10% (high) for all the cells present in the stromal or in the intratumoral compartment. Staining intensity was similar in all the tumors. Staining for PD-L1 was considered positive when >5% of the tumor cells presented membranous and/or cytoplasmic staining. In the TMA blocks, each tumor was represented by three 1-mm^2^ cores, and the one with the highest score was used. TMIT subtypes were defined by the combination of presence of intratumoral CD8^+^ TIL (CD8^low^, either 0% or 1–10%; and CD8^high^, >10%) and PD-L1-stained tumor cells (negative, <5%; and positive, >5%). TMIT I was defined as CD8^+^ TIL^high^/PD-L1^pos^; II—as CD8^+^ TIL^low^/PD-L1^neg^; III—as CD8^+^ TIL^low^/PD-L1^pos^; and IV—as CD8^+^ TIL^high^/PD-L1^neg^.

### 4.3. Statistical Analysis

Correlations between the results of immunohistochemistry and clinicopathological variables were analyzed using SPSS 15.0 for Windows (SPSS Inc., Chicago, IL, USA) using Pearson’s chi-squared test and Fisher’s exact test. Univariate Kaplan–Meier analysis was performed for the estimation of overall and disease-specific survival, comparing distributions using the Mantel–Cox logrank test. Values of *p* <0.05 were considered significant.

## Figures and Tables

**Figure 1 ijms-22-06926-f001:**
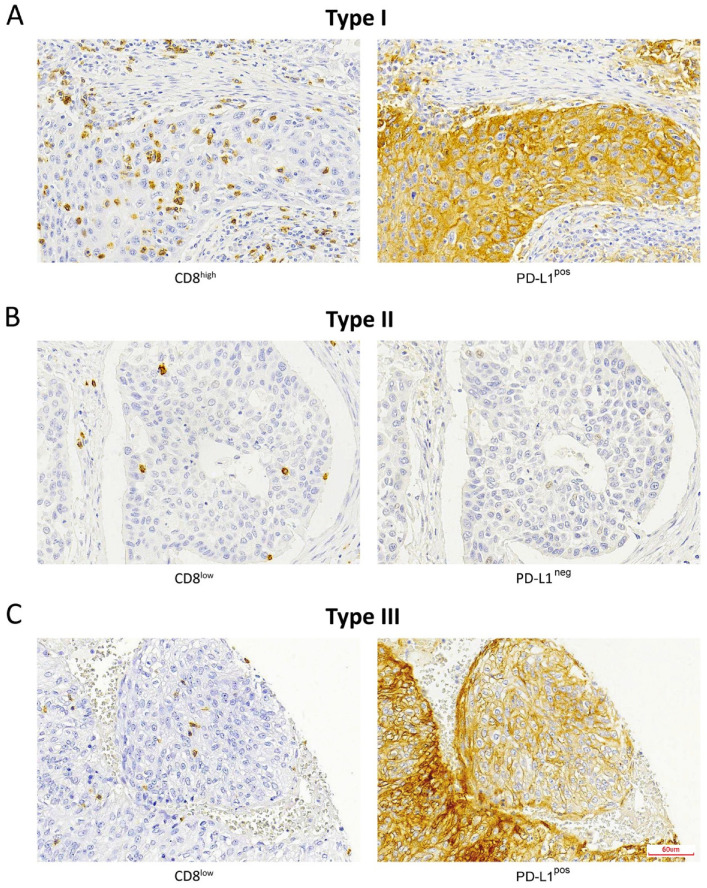
Immunohistochemical staining of CD8^+^ TILs and PD-L1-expressing tumor cells according to the TMIT classification: (**A**) TMIT I, (**B**) TMIT II, (**C**) TMIT III. TMIT IV was not observed. Magnification, 200×.

**Figure 2 ijms-22-06926-f002:**
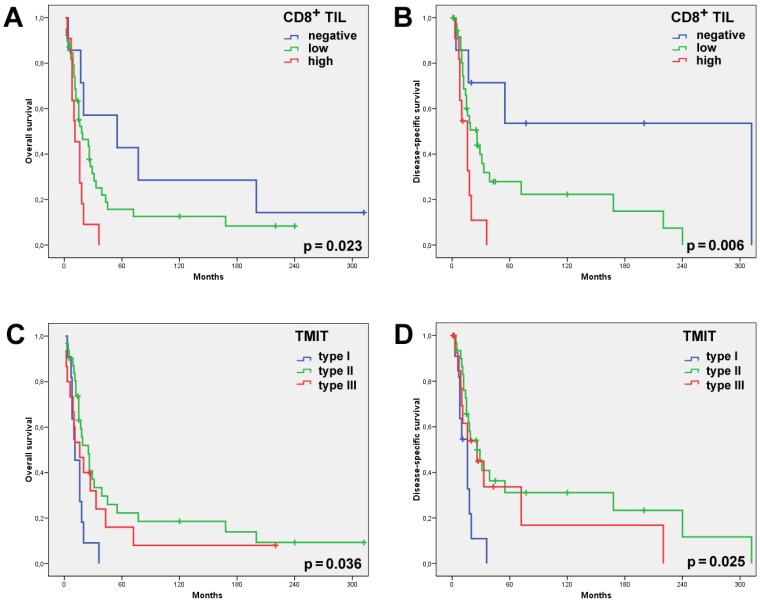
Univariate Kaplan–Meier analysis of all the 57 SNSCCs showed the CD8^high^ cases and, to a lesser extent, the CD8^low^ cases to have had a significantly worse overall (**A**) and disease-specific (**B**) survival than the cases with absent CD8^+^ TILs. Univariate Kaplan–Meier analysis of the overall (**C**) and disease-specific (**D**) survival according to the TMIT classification showed the TMIT I (CD8^high^/PD-L1^pos^) cases carried a significantly worse outcome.

**Figure 3 ijms-22-06926-f003:**
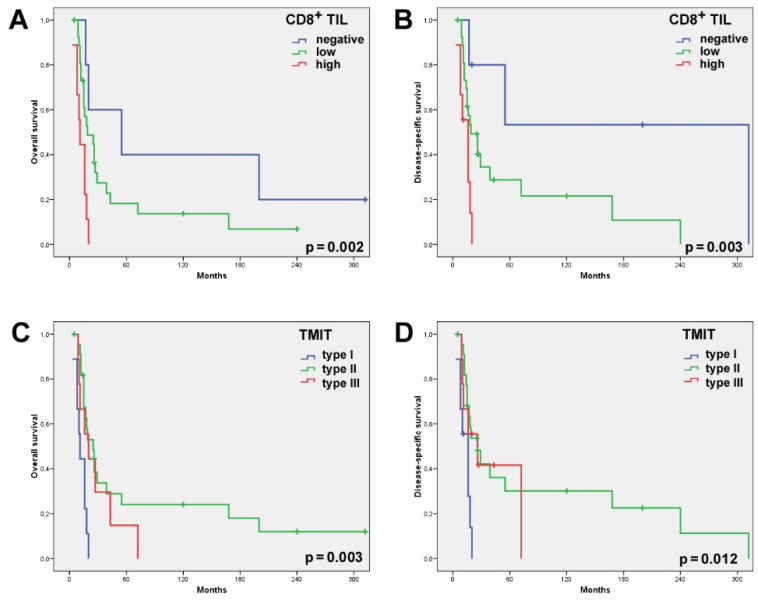
Univariate Kaplan–Meier analysis of the 41 SNSCC patients who had received radiotherapy showed the CD8^high^ cases and, to a lesser extent, the CD8^low^ cases to have a significantly worse overall (**A**) and disease-specific (**B**) survival than the cases with absent CD8^+^ TILs. Univariate Kaplan–Meier analysis of the overall (**C**) and disease-specific (**D**) survival according to the TMIT classification showed the TMIT I (CD8^high^/PD-L1^pos^) cases carried a significantly worse outcome.

**Table 1 ijms-22-06926-t001:** The relation of intratumoral and stromal expression of CD8^+^ TILs and TMIT to clinicopathological characteristics.

Clinical Characteristics	All	CD8^+^ TILs, Intratumoral				CD8^+^ TILs, Stromal				TMIT			
		0%	1–10%	>10%	*p*-value	0%	1–10%	>10%	*p*-value	I	II	III	*p*-value
All	57	7	39	11		1	32	24		11	31	15	
Gender					0.244				0.548				0.555
Female	20 (35)	4 (57)	11 (28)	5 (45)		0 (0)	10 (31)	10 (42)		5 (45)	9 (29)	6 (40)	
Male	37(65)	3 (43)	28 (72)	6 (55)		1 (100)	22 (69)	14 (58)		6 (55)	22 (71)	9 (60)	
Tumor site					0.117				0.497				0.282
Maxillary sinus	49 (86)	7 (100)	31 (80)	11 (100)		1 (100)	26 (81)	22 (92)		11 (100)	25 (81)	13 (87)	
Ethmoid sinus	8 (14)	0 (0)	8 (20)	0 (0)		0 (0)	6 (19)	2 (8)		0 (0)	6 (19)	2 (13)	
Disease stage					0.534				0.802				0.704
I	1 (2)	0 (0)	1 (3)	0 (0)		0 (0)	1 (3)	0 (0)		0 (0)	1 (3)	0 (0)	
II	3 (5)	0 (0)	3 (8)	0 (0)		0 (0)	2 (6)	1 (4)		0 (0)	3 (10)	0 (0)	
III	17 (30)	4 (57)	11 (28)	2 (18)		0 (0)	11 (34)	6 (25)		2 (18)	10 (32)	5 (33)	
IVa	25 (44)	3 (43)	15 (38)	7 (64)		1 (100)	14 (44)	10 (42)		7 (64)	11 (36)	7 (47)	
IVb	11 (19)	0 (0)	9 (23)	2 (18)		0	4 (13)	7 (29)		2 (18)	6 (19)	3 (20)	
Differentiation					0.828				0.525				0.712
Good	22 (39)	4 (57)	14 (36)	4 (36)		1 (100)	10 (31)	11 (46)		4 (36)	14 (45)	4 (27)	
Moderate	13 (22)	1 (14)	10 (26)	2 (18)		0 (0)	9 (28)	4 (17)		2 (18)	6 (19)	5 (33)	
Poor	22 (39)	2 (29)	15 (38)	5 (46)		0 (0)	13 (41)	9 (37)		5 (46)	11 (36)	6 (40)	
Recurrence					0.084				0.564				0.087
No	13 (23)	1 (14)	12 (31)	0 (0)		0 (0)	6 (19)	7 (29)		0 (0)	10 (32)	3 (20)	
Yes	44 (77)	6 (86)	27 (69)	11 (100)		1 (100)	26 (81)	17 (71)		11 (100)	21 (68)	12 (80)	
Metastasis					0.372				0.931				0.569
No	50 (88)	5 (71)	35 (90)	10 (91)		1 (100)	28 (88)	21 (87)		10 (91)	28 (90)	12 (80)	
Yes	7 (12)	2 (29)	4 (10)	1 (9)		0 (0)	4 (12)	3 (13)		1 (9)	3 (10)	3 (20)	
Patient status					0.449				0.790				0.252
Alive	8 (14)	1(14)	6 (15)	1 (9)		0 (0)	4 (12)	4 (17)		1 (9)	6 (19)	1 (6)	
Died of the disease	41 (72)	4 (57)	27 (70)	10 (91)		1 (100)	22 (69)	18 (75)		10 (91)	21 (68)	10 (67)	
Died of other causes	8 (14)	2 (29)	6 (15)	0 (0)		0 (0)	6 (19)	2 (8)		0 (0)	4 (13)	4 (27)	

**Table 2 ijms-22-06926-t002:** Correlation between intratumoral CD8^+^ TILs and PD-L1-expressing tumor cells.

	Score	PD-L1-Expressing Tumor Cells		
		Negative	Positive	*p*-value
CD8^+^ TILs	Negative (0%)	6	1	
	Low (1–10%)	25	14	0.000
	High (>10%)	0	11	

## Data Availability

The data presented in this study are available on request from the corresponding author.

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
