# Peer review of "Prognostic and Therapeutic Implications of Immune Classification by CD8+ Tumor-Infiltrating Lymphocytes and PD-L1 Expression in Sinonasal Squamous Cell Carcinoma"

_ijms, 2021, doi:10.3390/ijms22136926_

Round 1
Reviewer 1 Report
In this manuscript, the authors describe CD8+ TILs and PD-L1 status in a series of 57 sinonasal squamous cell carcinomas. They report a worse prognosis in patients harboring both CD8+ infiltrate and PD-L1 positivity. They conclude that these patients may benefit from anti-PD1 agents.
The article reads well. The study has potential clinical implications, and definitely deserves to be published.
Nevertheless, I have a few questions/suggestions:
M and M:
- Could the authors be more specific about PD-L1 and CD8+ TILs assessment? How was the cell count performed exactly? How were the fields determined: hotspots, random selection? The authors state that 3 independent observers evaluated the slides; it could be interesting to tell if there were interobservers discrepancies or if all observers obtained the same results.
- Table 1 displays important results of the study, and it seems awkward to show it in the M and M section; maybe the authors could describe the clinical characteristics of the patients at the beginning of the results section, so table 1 could appear in this section?
- The authors should specify whether they used univariate or multivariate analyses for OS and DSS.
- It’s just a detail, but were the slides really evaluated in a double-blind manner?
Results:
- For survival analyses, the authors should specify whether they used univariate or multivariate analyses.
- For survival analyses (“CD8+ TILs and correlation with clinical and survival data”, “TMIT and correlation with clinical and survival data” and figure 2): how was the statistical significance calculated? There are 3 groups on each survival curve (CD8+ negative/low/high and TMIT I/II/III), how should “p=0.023”… be interpreted?
- Could the authors elaborate on the results of double IF staining? The colocalization is not obvious on figure 3, would it be possible to describe it more precisely, and to provide a quantitative assessment of TILs and PD-L1+ cell colocalization? If these results are not available, the other option would be to skip this experiment.
Discussion:
- Major point: the interpretation of survival analyses is very difficult in retrospective series with a relatively low number of patients (rare tumors): tumor stage and treatment strategy may be very different from one patient to another. I would suggest to clearly state in the discussion section and in the abstract that all analyses are univariate (I guess), and that there are potential biases related to the fact that the series is retrospective and heterogeneous. The prognosis analysis should therefore be considered with caution. As pointed by the authors, this would also explain the discrepancies with the series of Quan et al (2019).
Abstract
- The sentence “Both intratumoural CD8+ TILs and PD-L1 positivity were related to worse survival” is not very clear, and suggests that the authors have assessed the association between PD-L1 positivity and survival, which is not the case in this manuscript. As TMIT analysis is an important part of the paper, I would suggest to describe the methods and results of TMIT assessment in the abstract.
- The sentence “we analysed CD8+ tumour-infilitrating lymphocytes (TILs) and PD-L1 16 expression as biomarkers for immunotherapy” suggests that response to immunotherapy has been assessed in patients with different TILs / PD-L1 status, which is not the case. The authors could rephrase to something like “As immunotherapy may be a new therapeutic option, we investigated TILs and PD-L1 status in a series of 57 SNSCC”.
Author Response
We would like to thank the reviewer for the valuable comments that have given the manuscript more clarity.
Review Report Form 1
Open Review
(x) I would not like to sign my review report
( ) I would like to sign my review report
English language and style
( ) Extensive editing of English language and style required
( ) Moderate English changes required
( ) English language and style are fine/minor spell check required
(x) I don't feel qualified to judge about the English language and style
Yes Can be improved Must be improved Not applicable
Does the introduction provide sufficient background and include all relevant references?
(x) ( ) ( ) ( )
Is the research design appropriate?
(x) ( ) ( ) ( )
Are the methods adequately described?
( ) (x) ( ) ( )
Are the results clearly presented?
( ) (x) ( ) ( )
Are the conclusions supported by the results?
(x) ( ) ( ) ( )
Comments and Suggestions for Authors
In this manuscript, the authors describe CD8+ TILs and PD-L1 status in a series of 57 sinonasal squamous cell carcinomas. They report a worse prognosis in patients harboring both CD8+ infiltrate and PD-L1 positivity. They conclude that these patients may benefit from anti-PD1 agents. The article reads well. The study has potential clinical implications, and definitely deserves to be published. Nevertheless, I have a few questions/suggestions:
M and M:
Could the authors be more specific about PD-L1 and CD8+ TILs assessment? How was the cell count performed exactly? How were the fields determined: hotspots, random selection? The authors state that 3 independent observers evaluated the slides; it could be interesting to tell if there were interobservers discrepancies or if all observers obtained the same results.
--> PD-L1 staining was scored only on tumour cells and considered positive when 5% or more of all tumour cells were stained. CD8+ TILs were considered intratumoural if they were encountered within the tumour cell areas, and stromal if they were encountered within the adjacent stroma. The density was evaluated in a semi-quantitative manner as the percentage of area (either tumoural or stromal) occupied by these lymphocytes. In the TMA, each tumor was represented by three cilinders and we used the highest score found in one of the cilinders. So yes, this was a kind of hotspot approach. Indeed the word double-blind was not chosen correctly; three authors (CR, SR and BV) did this evaluation and they were in agreement in the majority of cases (neither CD8 nor PD-L1 are difficult stainings to score), in those tumors where there was discrepancy, this was solved afterward by looking together using a multihead microscope. We made changes to this methods section.
Table 1 displays important results of the study, and it seems awkward to show it in the M and M section; maybe the authors could describe the clinical characteristics of the patients at the beginning of the results section, so table 1 could appear in this section?
--> The description of the patients and table 1 have been moved to the results section. We added one more follow-up information stating the 5-year disease-specific survival.
The authors should specify whether they used univariate or multivariate analyses for OS and DSS.
--> All statistical evaluations were univariate, we made this more clear in the text.
It’s just a detail, but were the slides really evaluated in a double-blind manner?
--> Please see our answer above.
Results:
For survival analyses, the authors should specify whether they used univariate or multivariate analyses.
--> Indeed we had not clarified that our survival analysis was done only in a univariate way. This is now added in the Results section.
For survival analyses (“CD8+ TILs and correlation with clinical and survival data”, “TMIT and correlation with clinical and survival data” and figure 2): how was the statistical significance calculated? There are 3 groups on each survival curve (CD8+ negative/low/high and TMIT I/II/III), how should “p=0.023”… be interpreted?
--> The statistical significance was calculated by Mantel Cox log rank test, for both CD8+ TILs and TMIT comparing the three different groups. The p-value indicates that the three groups have significantly different survival rates, so CD8-high cases and to a lesser extent also CD8-low cases have worse survival than cases with absence of CD8+ TILs. We added text to the figure legend to clarify.
Could the authors elaborate on the results of double IF staining? The colocalization is not obvious on figure 3, would it be possible to describe it more precisely, and to provide a quantitative assessment of TILs and PD-L1+ cell colocalization? If these results are not available, the other option would be to skip this experiment.
--> We should not have used the term colocalization, that would be when the same cell or the same cell compartment expresses two different proteins. What we wanted to illustrate was the observation that PD-L1 expressing tumour cells and CD8-positive T-cells are neighbouring, as possible indication of immune checkpoint interaction between the two. The reviewer is correct is saying that this is not a quantitative assessment and we agree to delete this experiment and the figure from the manuscript. What remains is the statistically significant quantitative assessment of co-occurrence of tumoural PD-L1 positivity and CD8+ TILs (table 2).
Discussion:
Major point: the interpretation of survival analyses is very difficult in retrospective series with a relatively low number of patients (rare tumors): tumor stage and treatment strategy may be very different from one patient to another. I would suggest to clearly state in the discussion section and in the abstract that all analyses are univariate (I guess), and that there are potential biases related to the fact that the series is retrospective and heterogeneous. The prognosis analysis should therefore be considered with caution. As pointed by the authors, this would also explain the discrepancies with the series of Quan et al (2019).
--> This is true. In the Discussion we have referred to differences in tumour stage and treatment strategy (our series are relatively higher stage and with a lower % of patients treated with radiotherapy and none with chemotherapy). We now added a sentence to more clearly state that the low number of patients also make that our univariate survival analysis must be considered with caution.
Abstract
The sentence “Both intratumoural CD8+ TILs and PD-L1 positivity were related to worse survival” is not very clear, and suggests that the authors have assessed the association between PD-L1 positivity and survival, which is not the case in this manuscript. As TMIT analysis is an important part of the paper, I would suggest to describe the methods and results of TMIT assessment in the abstract.
--> Thanks for this comment, indeed this sentence does not express correctly what was done. It should state that both intratumoural CD8+ TILs and TMIT type I correlated to worse survival. Accordingly, we have made changes to the abstract.
The sentence “we analysed CD8+ tumour-infilitrating lymphocytes (TILs) and PD-L1 16 expression as biomarkers for immunotherapy” suggests that response to immunotherapy has been assessed in patients with different TILs / PD-L1 status, which is not the case. The authors could rephrase to something like “As immunotherapy may be a new therapeutic option, we investigated TILs and PD-L1 status in a series of 57 SNSCC”.
--> We agree with your comment. Ours is a retrospectivey study, we did not analyse CD8+ TILs and TMIT in relation to immunotherapy-response in these patients. We have changed the sentence.
Reviewer 2 Report
In this manuscript (ijms-1255515), Garcìa-Marìn and co-authors analyzed the CD8+ tumor-infiltrating lymphocytes (TILs) and PD-L1 expression on tumor cells as biomarker for immunotherapy in sinonasal squamous cell carcinoma (SNSCC) and correlated their results to clinical and survival data. They showed that SNSCC is an immunogenic tumor and that CD8+ TILs high expression and PD-L1 positivity correlate to worse survival, which in turn may respond to therapy with immune checkpoint inhibitors.
The issue is interesting, giving the actual attention to immunotherapy. However, the work is weak and several issue should be addressed:
- It is important to have an overview of the clinical characteristics of the patients enrolled in the study. So, Table 1 should be added as results instead that in materials and methods section. Moreover, it is not always clear what the numbers are referred to. The authors should indicate what the numbers means in an appropriate table legend to render it clearer to the readers. The word “All” should be aligned to the column to which is referred.
- How can they exclude that radiotherapy did not influence their results on survival rate?
- Why the authors evaluate only the presences of CD8+ tumor-infiltrating lymphocytes and not consider stromal ones? The authors should show both the intratumoral and stromal CD8+ or CD8- in table 1 and the related survival rate and discuss their results.
- It is not clear what the authors wanted to show with the immunofluorescence in figure 3. Moreover, the information obtained is not sufficient to support their conclusion.
Author Response
We would like to thank the reviewer for the valuable comments that have given the manuscript more clarity.
Review Report Form 2
Open Review
(x) I would not like to sign my review report
( ) I would like to sign my review report
English language and style
( ) Extensive editing of English language and style required
(x) Moderate English changes required
( ) English language and style are fine/minor spell check required
( ) I don't feel qualified to judge about the English language and style
Yes Can be improved Must be improved Not applicable
Does the introduction provide sufficient background and include all relevant references?
(x) ( ) ( ) ( )
Is the research design appropriate?
( ) ( ) (x) ( )
Are the methods adequately described?
( ) (x) ( ) ( )
Are the results clearly presented?
( ) ( ) (x) ( )
Are the conclusions supported by the results?
( ) ( ) (x) ( )
Comments and Suggestions for Authors
In this manuscript (ijms-1255515), Garcìa-Marìn and co-authors analyzed the CD8+ tumor-infiltrating lymphocytes (TILs) and PD-L1 expression on tumor cells as biomarker for immunotherapy in sinonasal squamous cell carcinoma (SNSCC) and correlated their results to clinical and survival data. They showed that SNSCC is an immunogenic tumor and that CD8+ TILs high expression and PD-L1 positivity correlate to worse survival, which in turn may respond to therapy with immune checkpoint inhibitors.
The issue is interesting, giving the actual attention to immunotherapy. However, the work is weak and several issue should be addressed:
It is important to have an overview of the clinical characteristics of the patients enrolled in the study. So, Table 1 should be added as results instead that in materials and methods section. Moreover, it is not always clear what the numbers are referred to. The authors should indicate what the numbers means in an appropriate table legend to render it clearer to the readers. The word “All” should be aligned to the column to which is referred.
--> This comment agrees with that of reviewer 1. The description of the patients and table 1 have been moved to the results section. Also, we have moved the word ''All'' to its right place and we added a legend to the table. The numbers in the colums first refer to the absolute number of tumours and then between brackets to the % of tumours. For example, at 0% CD8+ TILs, the total number of cases is 7, of which 4 (57%) are female and 3 (43%) are male. The significance p-value is the result of statistical analysis calculated by Pearson Chi2.
How can they exclude that radiotherapy did not influence their results on survival rate?
--> Thank you for this comment. In Discussion we suggested that discrepancies in the relation of CD8+ TILs and TMIT to survival may be due to differences in tumour stage and received treatments. In our series only 72% (41/57) of patients received radiotherapy. We had not analysed survival in only the cases with radiotherapy because of the even lower number of cases (41 in stead of 57). However, we now see that the relation of CD8+ TILs and TMIT to overall and disease-specific survival in only the 41 patients that received radiotherapy becomes statistically more significant: CD8+ TILs vs overall survival in 57 cases was p=0.023 and in the 41 cases p=0.002, and TMIT vs overall survival in 57 cases was p=0.036 and in the 41 cases p=0.003. So this shows that radiotherapy does influence our results on survival rate. As the Kaplan Meier curves of this analysis of 41 cases look very similar to the ones given in figure 2 reflecting all 57 cases, we decided to leave the figure as it is. However, we added this extra information to the Results sections.
Why the authors evaluate only the presences of CD8+ tumor-infiltrating lymphocytes and not consider stromal ones? The authors should show both the intratumoral and stromal CD8+ or CD8- in table 1 and the related survival rate and discuss their results.
--> Our motive no not evaluate stromal CD8+ TILs was the fact that they were present in 98% (56/57) of the tumours, which would make looking for relations to clinical data not interesting. However, there were differences between high and low presence, respectively 42% and 56% of cases. Therefore, we have done as the reviewers suggests and expanded table 1 complete with Pearson Chi2 analysis of the data in relation to clinical features. In Results, it was already stated that stromal CD8+ TILs did not correlate with survival, and we now added the log rank p-values. In Discussion page 7 we added this sentence: 'This correlation was only significant with intratumoural presence and not with stromal presence, suggesting that actual contact between tumour cells and CD8+ TILs is important.'
It is not clear what the authors wanted to show with the immunofluorescence in figure 3. Moreover, the information obtained is not sufficient to support their conclusion.
--> Please see our answer to reviewer 1; we agree to delete this experiment and the figure from the manuscript.
Round 2
Reviewer 2 Report
The authors revised the manuscript accordingly to reviewers’ corrections and suggestions and the overall presentation of the manuscript is improved. However, some issue should be addressed before the publication to IJMS.
- I really appreciate the analysis of the influence of radiotherapy on survival rate. However, a discussion of this new results is still missing.
- In the result 2, a representation of the survival rates should be added to render they results clearer to the reader.
- The correlation between the presence of intratumoral CD8+ TILs and PD-L1 expression is not sufficient to support their conclusion. The authors could not give evidence of an interaction between this cell types. They can only speculate.
Author Response
Open Review
(x) I would not like to sign my review report
( ) I would like to sign my review report
English language and style
( ) Extensive editing of English language and style required
( ) Moderate English changes required
(x) English language and style are fine/minor spell check required
( ) I don't feel qualified to judge about the English language and style
Yes Can be improved Must be improved Not applicable
Does the introduction provide sufficient background and include all relevant references?
(x) ( ) ( ) ( )
Is the research design appropriate?
( ) (x) ( ) ( )
Are the methods adequately described?
(x) ( ) ( ) ( )
Are the results clearly presented?
( ) ( ) (x) ( )
Are the conclusions supported by the results?
( ) ( ) (x) ( )
The authors revised the manuscript accordingly to reviewers’ corrections and suggestions and the overall presentation of the manuscript is improved. However, some issue should be addressed before the publication to IJMS.
I really appreciate the analysis of the influence of radiotherapy on survival rate. However, a discussion of this new results is still missing.
--> Yes we had only added the information to the Results section. Now we added Figure 3 that shows the survival in relation to CD8+ TILs and TMIT of only the 41 patients with radiotherapy, and in Discussion we added the following sentence: '' To take into consideration the influence of radiotherapy on survival, we carried out a separate univariate survival analysis of only the 41 patients that had received radiotherapy after radical surgery. This produced a even stronger relation of CD8+ TILs and TMIT type I to worse overall and disease-specific survival (Figure 3) and therefore does not offer an explanation for the discrepancy with the main body of literature.
In the result 2, a representation of the survival rates should be added to render they results clearer to the reader.
--> We assume this comment also refers to the influence of radiotherapy on survival. We have now added a Figure 3 giving the Kaplan Meier overal and disease-specific survival curves according to CD8+ TILs and to TMIT with only the 41 patients that had received radiotherapy.
The correlation between the presence of intratumoral CD8+ TILs and PD-L1 expression is not sufficient to support their conclusion. The authors could not give evidence of an interaction between this cell types. They can only speculate.
--> Yes it is only a speculation, we really can't make statement on the actual interaction. We decided to delete this sentence: ''Looking for a possible explanation, we investigated if CD8+ TILs perhaps do not interact with PD-L1 expressing tumour cells. However, we found that they significantly co-occurred (p=0.000), which suggests that there is interaction between the two.'' In the following sentence we kept the speculation that the CD8+ TILs may be dysfunctional exhausted cells, but deleted ''probably''.
We read the text carefully again and corrected a number of typographic and other errors.